# Relationship between Fear of COVID-19, Intolerance of Uncertainty, and Coping Strategies on University Students’ Mental Health

**DOI:** 10.3390/nu15234938

**Published:** 2023-11-28

**Authors:** Elodie Charbonnier, Lucile Montalescot, Cécile Puechlong, Aurélie Goncalves, Sarah Le Vigouroux

**Affiliations:** UNIV. NIMES, APSY-V, F-30021 Nîmes Cedex 1, France; lucile.montalescot@unimes.fr (L.M.); cecile.puechlong@gmail.com (C.P.); aurelie.goncalves@unimes.fr (A.G.); sarah.le_vigouroux_nicolas@unimes.fr (S.L.V.)

**Keywords:** anxiety, depression, eating disorder, prospective, inhibitory, intolerance of uncertainty

## Abstract

Background: the mental health of students was particularly affected by the COVID-19 pandemic. The present study therefore examined the relationships between anxiety and depressive symptoms, eating-related problems, coping, fear of COVID-19, and intolerance of uncertainty. Methods: 2139 French students of 54 universities were recruited in the different regions of France during a French lockdown (between 21 April and 3 May 2021). Six variables were measured: fear of COVID-19, intolerance of uncertainty, coping, anxiety and depressive symptoms, and eating-related problems. To explore the directions of the relationships between our variables of interest, we calculated a directed acyclic graph. Results: our data highlighted the central roles of intolerance of uncertainty in students’ anxiety and depressive symptoms, and the direct role of prospective intolerance of uncertainty on eating-related problems. Conclusions: these findings indicate that intolerance of uncertainty should be targeted by interventions designed to help students with high levels of anxiety, depressive symptoms, and/or eating-related problems.

## 1. Introduction

The COVID-19 pandemic had a very negative impact on people’s mental health [1], with university students being particularly badly affected [2,3,4]. Several studies have revealed high levels of eating disorders [5,6], anxiety [3,7,8] and depressive symptoms [9,10,11,12] among university students. During the first two lockdowns, 30–37% of French university students exhibited possible depressive symptoms, and 45–48% exhibited possible anxiety [13]. These data are very alarming, especially as students had already been identified as being vulnerable to these mental health problems prior to the pandemic [14,15,16], and studies tracking the long-term consequences of previous pandemics have suggested that psychological problems may last for years (e.g., [17,18,19]).

The COVID-19 pandemic has been associated with *fear of disease*, which can lead to emotional and behavioral problems [20]. Although fear can be regarded as a negative emotion, it is also necessary for survival [21]. During a pandemic such as COVID-19, fear is a normal and realistic response, and is one of the reasons why people comply with the various health recommendations [22]. Nevertheless, fear of COVID-19 triggered many concerns about food, including the possibility that food might be a vector for contagion, and this may have led to the emergence of restrictive eating behaviors [23]. For example, the authors of [5] found that the more individuals feared COVID-19, the more they engaged in restrictive eating behaviors. More generally, fear of COVID-19 has been positively associated with psychological distress [24,25,26]. In particular, a meta-analysis [20] showed that fear of COVID-19 was positively and strongly related to anxiety, and positively and moderately related to depressive symptoms and stress.

Beyond the fear of COVID-19, the deterioration observed in students’ mental health can partly be explained by the high levels of stress generated by the pandemic [27,28]. Stress occurs when individuals perceive the demands of the environment to be a threat to their wellbeing [29], and pandemics are particularly threatening and therefore stressful [20]. During the COVID-19 pandemic, this stress particularly affected students, partly because many found themselves isolated, away from family and friends [30], with many new challenges to contend with, such as online courses and distance exams [8,27].

Prior to the pandemic, the association between stress and mental health had been shown to be largely dependent on coping strategies [31,32]. Coping is a process that allows us to adapt to stressful situations [29]. Adaptive strategies (e.g., cognitive restructuring, problem solving) refer to strategies that allow for a favorable outcome to the situation, reduce stress in the long term, and are therefore protective factors for mental health [33,34]. For example, the more students use positive reappraisal and/or problem solving, the less likely they are to experience an increase in depressive symptoms after moderate and recurrent stressors (e.g., receiving a disappointing exam grade; [35]). By contrast, maladaptive strategies (e.g., avoidance, isolation) refer to rigid and ineffectual behaviors that do not improve the situation, increase stress, and are therefore risk factors for mental health [33,34]. 

The central role of coping strategies in students’ mental health was demonstrated during the COVID-19 pandemic, for both anxiety and depressive symptoms [4,13], as well as eating disorders [36]. In addition, the authors of [37] found that after 6 months of the pandemic, medical students made less use of adaptive strategies (e.g., positive reframing, acceptance) and more use of maladaptive strategies (e.g., denial, emotional venting, substance use, activity sup-pression, and self-blame) than before. The changes observed in students’ coping may be explained in part by the fact that some of the strategies they had previously used were no longer available (especially during lockdown), or else were less relevant in the face of the uncontrollable situations generated by COVID-19. The COVID-19 pandemic exposed populations to new, largely uncontrollable, and unprecedentedly stressful experiences [38]. Given that uncertainty was also greatly exacerbated during the pandemic [39], especially among students [40], it seems worthwhile exploring their *intolerance of uncertainty* (IU), in order to better understand their choice of coping strategies, as well as their deteriorating mental health.

IU can be defined as the excessive tendency of an individual to view the possible occurrence of a negative event as unacceptable, however low the probability [41]. It is regarded as a cognitive bias that influences the way people perceive, interpret, and react to uncertain situations [42]. Thus, the higher people’s levels of IU, the more likely they are to perceive ambiguous information as threatening [43]. IU levels were found to be very high during the pandemic [39], especially among students [40]. Several studies found moderate associations between IU and mental health, with high IU levels being associated with poorer mental health, including anxiety [42], depressive symptoms [44], and eating disorders [45]. High levels of IU reinforce the relationship between anxiety and everyday stressors [46], and predict anxiety levels during exceptional circumstances (e.g., the H1N1 pandemic in 2009; [47]). Furthermore, IU is associated with the use of particular coping strategies, such as excessive information seeking and vigilant coping strategies (i.e., increased processing of threatening information; [48]), or avoidance, inaction, and inhibition strategies [49,50]. High levels of IU may lead individuals to make more rapid decisions and act impulsively, in order to alleviate their distress in stressful situations [49] and thus reduce their uncertainty more quickly [51].

Some studies have suggested that the links between IU, mental health, and coping differ according to which IU dimensions are considered (prospective or inhibitory). *Prospective IU* corresponds to a need to be able to anticipate future events, and translates into behaviors aimed at reducing uncertainty. When it is high, it can lead individuals to engage in information gathering and action planning strategies to reduce uncertainty [48]. *Inhibitory IU* corresponds to impaired functioning (e.g., rumination, delayed decision making) resulting from uncertainty [52] and promotes avoidance strategies [50]. Prospective IU appears to be correlated more with worry and symptoms of obsessive compulsive disorder, whereas inhibitory IU is more strongly associated with social anxiety, panic disorder, agoraphobia, posttraumatic stress disorder, and depressive symptoms [52,53,54,55]. Both dimensions of IU are related to students’ anxiety and depressive symptoms, with stronger relationships for prospective IU [56].

### Objective and Hypotheses

Given that students were particularly affected by the COVID-19 pandemic (high levels of anxiety, depressive symptoms, and eating disorders), the purpose of the present study was to identify the factors involved in the deterioration of students’ mental health. More specifically, the aim of this study was to identify, in an innovative way, the direct and indirect relationships between the various predictors identified in the pre-COVID literature and different mental health indicators, in a particular context of stress and uncertainty, namely a period of lockdown. In the light of previous findings [4,13], we hypothesized that the more students used maladaptive strategies and the less they used adaptive strategies, the more severe their anxiety, depressive symptoms, and eating-related problems (Hypothesis 1). Given the links between IU, coping, and mental health established prior to the pandemic [44,46,49], we also hypothesized that the higher students’ levels of IU, the greater their use of maladaptive coping strategies (Hypothesis 2), and the more severe their anxiety, depressive symptoms, and eating-related problems (Hypothesis 3). Finally, we hypothesized that the more students feared COVID-19, the greater their use of maladaptive strategies (Hypothesis 4), and the more severe their symptoms (Hypothesis 5). Finally, in a more exploratory approach (Hypothesis 6), we hypothesized a direct relationship between clinical issues (i.e., anxiety, depressive symptoms, and eating-related problems), and dispositional dimensions (i.e., IU), as well as an indirect relationship between these variables, via coping strategies and fear of COVID-19.

## 2. Materials and Methods

### 2.1. Participants and Procedure

To recruit our participants, a link to our online survey designed with Qualtrics software (Qualtrics, Provo, UT, USA, https://www.qualtrics.com, License date: from November 2020 to November 2021) was sent by e-mail to teachers in French universities, who forwarded it to their students. The link to the survey was also distributed via students’ social media (e.g., Facebook, Discord, Twitter, Instagram). Data were collected anonymously between 21 April and 3 May 2021. This period was chosen because it was a lockdown period, the first of 2021, and the third lockdown period in France since the start of the pandemic. This period was particularly critical for students. Indeed, for the second year running, they had to complete their university year at a distance (courses and exams), with a curfew, and without the possibility of going to leisure facilities (restaurants, cinemas, etc.). To ensure the reliability of the answers, Qualtrics software ensures that a participant cannot respond twice to the link with the same IP address. In addition, all incomplete questionnaires and/or with aberrant answers (e.g., with identical answers to all questions) have been deleted.

To determine the required number of participants, we ran a power test using the pwr.r.test function of the pwr package [57]. Based on the strength of the relationships found in previous studies, the power test (*r* = 0.20, *p* = 0.05, power = 0.80) indicated a sample size of 193 participants. To ensure relevant statistical power, we recruited 2139 French students (1612 women, 506 men, and 21 nonbinary) aged 18–66 years (*M* = 21.56, *SD* = 4.38). They came from 54 different universities across France, including the Universities of Paris City (21%) Nîmes (16%), Lorraine (14%), Montpellier (6%), and Strasbourg (5%). The sample’s descriptive data are set out in Table 1. 

### 2.2. Measures

*Fear of COVID* was assessed using a French translation of the Fear of COVID-19 Scale [58]. Each of its seven items (e.g., “I cannot sleep because I’m worrying about getting coronavirus-19”) is rated on a 5-point Likert scale ranging from 1 (strongly disagree) to 5 (strongly agree). The higher the score, the greater the fear of COVID-19. In our study, internal consistency for this scale was satisfactory (α = 0.87).

*IU* was assessed using a French translation of the Intolerance of Uncertainty Scale—Short Form [59]. This self-report scale measures responses to items about uncertainty, ambiguous situations, and the future. It has a two-factor structure [54]: prospective IU, which relates to anxiety and fear of future events (seven items, e.g., “One should always look ahead so as to avoid surprises”); and inhibitory IU, which impedes action (five items, e.g., “When it’s time to act, uncertainty paralyses me”). These 12 items are rated on a 5-point Likert scale ranging from 1 (not at all characteristic of me) to 5 (entirely characteristic of me). Higher scores reflect higher levels of IU. In our study, internal consistency was satisfactory for each of the two subscales (αProspective IU = 0.87, αInhibitory IU = 0.87).

*Coping strategies* were assessed using a French version of the situational version of the Brief-COPE [34]. This self-report scale assesses 14 coping strategies with 28 items: nine adaptive strategies (e.g., active coping, acceptance), and five maladaptive strategies (e.g., behavioral disengagement, substance use). For each, participants responded on a 4-point Likert scale from 0 (never) to 3 (always). The higher the score, the more the strategy is used. In this study, participants were instructed to refer to a stressful situation related to the lockdown. 

*Anxiety and depressive symptoms* were assessed using a French version of the Hospital Anxiety and Depression Scale [60]. This scale assesses the intensity of anxiety and depressive symptoms during the previous week. Each dimension is measured using seven items, with scores ranging from 0 to 21. The higher the score, the more severe the anxiety and/or depressive symptoms. More precisely, a score ≤ 7 indicates no symptoms, a score between 8 and 10 corresponds to possible symptoms, and a score ≥ 11 signifies probable symptoms. *Eating-related problems* were assessed using a French version of the Eating Attitudes Test-26 [61]. This 26-item scale measures symptoms and concerns characteristic of eating disorders. It can be used in both nonclinical and clinical settings that are not specifically focused on eating disorders. For each item, participants responded on a 6-point Likert scale from 1 (never) to 6 (always). A score ≥ 20 corresponds to a high level of concern about dieting, body weight, or problematic eating behaviors.

### 2.3. Compliance with Ethical Standards

This study was funded by “Agence Nationale de la recherche—ANR” (COV’Etu—ANR-21-COVR-0005). This study was an experiment in human and social sciences in the field of health. It was conducted in accordance with institutional and national ethical standards, and in accordance with the ethical standards as laid down in the Declaration of Helsinki (1964). Recruitment was based on voluntary participation, with no compensation. Participants signed an online informed consent form. They were informed that their information would remain anonymous and that their participation could be withdrawn at any time. At the start of the questionnaire, the names and contact details of the experimenters were specified, should participants wish to have more information about the research. All data were stored in secure environments. 

### 2.4. Statistical Analysis

First, we explored gender differences on the main variables of interest using ANOVA. Second, we estimated the correlations (Pearson’s) to highlighted the relationships between all our variables. Third, we estimate a directed acyclic graph (DAG) in order to explore the directions of relationship between our variables of interest with R (version 4.3.0, R core team, 2023) and bnlearn package [62]. For this purpose, we need a positive correlation matrix (we therefore inverted the adaptative coping score, in order to include it). Then, we estimated the structure of the network. We used the bootstrap function that computes the structural aspects of the network model by adding removing, and testing direction of edges, allowing to estimate the most optimum and parsimonious network of directed relations between the variables. This network is estimated with an iterative procedure of randomly restarting using 50 different random restarts to avoid local maxima, and 100 perturbations (i.e., attempts to insert, delete, or reverse an edge) for each restart. Finally, we tested the stability of the DAG obtained with a bootstrapping method (on 10,000 samples with replacement) [63].

## 3. Results

In preliminary analyses, we studied the effect of gender on our different variables (Appendix A). Although some statistically significant differences were present, the effect sizes of these differences were very small. More specifically, post hoc tests indicated that women experience slightly more fear of COVID, depressive symptoms and eating-related problems than men. In addition, it appears that non-binary people experienced slightly more anxiety symptoms than women, who also experienced slightly more than men. Furthermore, women and non-binary people reported slightly higher intolerance of uncertainty and maladaptive coping than men. Finally, there were no gender differences in adaptive coping.

Consistent with our hypotheses, our correlational analyses (Table 2) revealed significant associations between clinical outcomes (i.e., anxiety, depressive symptoms, and eating-related problems), coping strategies, IU, and fear of COVID-19. More specifically, maladaptive strategies were positively and strongly related to anxiety and depressive symptoms, and positively and weakly related to eating-related problems (Hypothesis 1). In addition, IU was positively and moderately related to maladaptive strategies (Hypothesis 2), as well as to anxiety and depressive symptoms, and weakly related to eating-related problems (Hypothesis 3). Finally, fear of COVID-19 was positively and moderately related to maladaptive strategies (Hypothesis 4), as well as to anxiety and depressive symptoms, and weakly related to eating-related problems (Hypothesis 5).

Figure 1 shows the graphical network between our variables of interest. Strong and positive edges are apparent between prospective and inhibitory IU nodes (Nodes 4 and 5), and between anxiety and depressive symptoms (Nodes 2 and 3). Moreover, anxiety is the most central variable in the network (Figure 2). The more anxiety students reported, the more they also reported depressive symptoms, fear of COVID-19 (Node 1), use of maladaptive coping strategies (Node 7), and inhibitory IU. 

To validate our exploratory hypothesis (Hypothesis 6), we used DAGs to test directed relationships between clinical issues (i.e., anxiety, depressive symptoms, and eating-related problems), dispositional dimensions (i.e., prospective and inhibitory IU), coping strategies (i.e., adaptive and maladaptive coping), and fear of COVID-19. As this approach requires unsigned (i.e., all positive) relations between the variables, we reversed the adaptive coping scores. Figure 3 shows the DAGs resulting from 10,000 bootstrapped samples. In both graphs (Figure 3), edges were retained because their strength was greater than the optimal cut point yielded by the method presented in [63]. The first graph (Figure 3; network structure) illustrated the importance of each edge to the overall network structure. The second graph (Figure 3; edge direction) illustrated the probability of the prediction being in the depicted direction, with thick arrows indicating high directional probabilities, and thin arrows low directional probabilities. 

More specifically, in the network structure graph (Figure 3), the thicker the edge, the more crucial it was to the model fit [64]. The edges that were most important to the network structure connected prospective and inhibitory IU and depressive symptoms to anxiety, with changes in the Bayesian information criterion (BIC) of −775.98 and −383.87. The edges least important to the network structure, meanwhile, connected prospective and inhibitory IU to fear of COVID, with BIC changes of −0.43 and 0.14. 

In the edge direction graph (Figure 3), edges signified directional probabilities, such that an edge was thicker if it pointed from one node to another in a greater proportion of the bootstrapped networks. The thickest edges connected prospective IU to eating-related problems (0.98, i.e., this edge pointed in that direction in 9800 of the 10,000 bootstrapped networks, and in the other direction in 200 of the 10,000 bootstrapped networks), and maladaptive coping to eating-related problems (0.90). The thinnest edges connected maladaptive coping to adaptive coping (0.51), and depressive symptoms to anxiety (0.54).

At the structural level, inhibitory IU sat at the top of the DAG, directly influencing anxiety, depressive symptoms, maladaptive coping, and prospective IU. In turn, the latter two variables directly influenced fear of COVID and eating attitudes, while depressive symptoms and maladaptive coping directly influenced adaptive coping.

## 4. Discussion

Several studies have shown that students have high levels of anxiety, depressive symptoms, and eating disorders [14,15,16], which were exacerbated by the COVID-19 pandemic [2,5]. Furthermore, the links between IU, coping, and mental health were established prior to the pandemic [44,46,49] but few studies have considered these factors during the COVID-19 pandemic, and none have studied them together. More specifically, the aim of this study was to identify the direct and indirect relationships between different factors identified in the pre-COVID literature (i.e., UI, coping), factors identified during COVID (i.e., fear COVID) and different mental health indicators (i.e., anxiety, depressive symptoms, and eating-related problems), and this in a particular context of stress and uncertainty (a lockdown period, with distance learning courses and exams). Our results highlighted the central roles of IU and, to a lesser extent, of coping in students’ anxiety and depressive symptoms, as well as in their eating-related problems.

First, consistent with the authors of [65]’s meta-analysis, our correlational analyses revealed that anxiety and depressive symptoms were mainly associated with IU (prospective and inhibitory—Hypothesis 3) and maladaptive strategies (Hypothesis 1). More innovatively, and in line with our Hypothesis 6, our DAG clarified the nature of these relationships, by highlighting the central role played by inhibitory IU in students’ anxiety and depressive symptoms during the COVID-19 pandemic, as it had done before [56]. This can be explained by the fact that inhibitory IU is strongly associated with different factors involved in anxiety and depressive symptoms, such as avoidance [50,66], rumination, and negative affect [56]. Beyond its major role in anxiety and depressive symptoms, our DAG also revealed that inhibitory IU was directly related to maladaptive strategies. This may be because the greater the inhibitory IU, the greater the cognitive avoidance [67]. Individuals may therefore have difficulty coping with challenging situations, such as those experienced during the pandemic, possibly leading to the use of maladaptive strategies [46,68,69]. Furthermore, given that uncertain situations are omnipresent in daily life and were exacerbated during the COVID-19 pandemic, the desire to strongly reduce uncertainty was probably often unattainable, leading to the use of maladaptive strategies such as denial and self-blame.

Second, our correlational analyses showed that eating-related problems were positively (albeit weakly) associated with IU and maladaptive strategies, consistent with previous studies (e.g., [65,70]) and our Hypotheses 1 and 2. In a more innovative manner, our DAG highlighted the direct role of prospective IU in students’ eating-related problems during the pandemic. Prospective IU is associated with a strong desire to predict future events, leading individuals to adopt strategies intended to reduce their uncertainty. Our results suggest that adopting dysfunctional eating behaviors may have been a strategy used by students to reduce their uncertainty during the pandemic. This hypothesis is supported by the fact that food restriction can reduce anorexic patients’ uncertainty, such as when they do not know the exact composition of a dish [71]. Although the links between eating-related problems and maladaptive coping have been widely demonstrated (see meta-analysis by [72]), little consideration has been given to the role of coping in specific contexts such as COVID-19. Our DAG demonstrated that eating problems during COVID-19 can be explained by the use of maladaptive strategies to cope with stressful situations related to the pandemic. 

Finally, contrary to our expectations and more precisely our Hypothesis 5, fear of COVID-19, did not emerge as an explanatory factor for eating-related problems, anxiety, or depressive symptoms among students during the COVID-19 pandemic. Although our correlational analyses revealed moderate associations between these variables, the DAG showed that fear of COVID-19 was explained directly by anxiety, and indirectly by depressive symptoms, and not the reverse. In addition, eating-related problems were independent of fear of COVID-19. These data lead us to moderate the conclusion of the authors of [20]’s meta-analysis stating that when fear of COVID-19 becomes dysfunctional, it can lead individuals to develop anxiety and depressive symptoms. One possible explanation for the impact of anxiety on fear of COVID-19 is that individuals with high levels of anxiety attend more quickly to threatening stimuli in their environment and allocate more attentional resources to them [73,74]. Thus, we can speculate that students with elevated levels of anxiety were more likely to exhibit heightened vigilance for COVID-19 and possible contagion, resulting in greater fear of COVID-19.

### Limitations and Perspectives

The present study also had several limitations. First, our results exclusively concerned French students, thus limiting their generalization. However, it is important to note that studies in other countries (1) also reported increased anxiety, depressive symptoms, and eating-related problems in university students during lockdown, and (2) linked UI and coping strategies to these symptoms. In the same vein, although we chose to target students because they are more vulnerable, there is every reason to believe that the relationships highlighted in this study could also be found in the general population. This suggests that our results may also be useful in other countries and population, although additional research is needed to consolidate these findings. Second, our data were collected during a specific context, i.e., lockdown. Consequently, these results should be viewed with caution as a number of variables related to this specific context could have had an impact on some of our scores and were not captured in this study (e.g., having been directly or indirectly affected by COVID). For example, it has been shown that university students with COVID-19 symptoms, and/or having parents with COVID-19 symptoms, had higher levels of anxiety and depressive symptoms [4]. In addition, lockdown was a period in which students were particularly vulnerable and exposed to much uncertainty. Although several studies have highlighted the lasting psychological effects of pandemics, it seems essential to rerun this protocol now that the epidemic is under control, particularly to consolidate the links between the two dimensions of IU, and anxiety, depressive symptoms and eating-related problems. Third, in order to clarify the associations between our variables of interest and to explore the directions of these relationships, we chose to use innovative statistical analyses (i.e., DAG). This approach yielded innovative results. However, although DAGs are very informative, as they represent an estimate of both the strength and direction of relationships between network variables, results on cross-sectional data should be viewed with caution. This type of analysis cannot replace longitudinal studies, which alone can reveal causal relationships. Moreover, DAGs do not allow retroactive relationships to be estimated. DAG estimates are, by definition, relationship-directed and acyclic. The low probability of most directions in the DAGs (e.g., inhibitory IU–anxiety; anxiety–depression) suggests that there were hidden feedback loops between these constructs. For example, inhibitory IU may trigger anxiety, and anxiety may in turn increase inhibitory IU.

Despite these limitations, the results of the present study provide new insights leading to clinical implications for the support of students with elevated levels of anxiety, depressive symptoms, and/or eating-related problems. They demonstrate that IU could constitutes a prime target for intervention, particularly inhibitory IU. Consistent with this, changes in IU scores have been shown to predict changes in mental health [75]. This is reinforced by the fact that IU is identified as a process that can be modified by therapeutic interventions, such as cognitive and behavioral therapies (e.g., [76]). This appears especially appropriate for students, as an 8-week online or face-to-face intervention conducted during the COVID-19 pandemic was shown to significantly reduce students’ IU scores [77].

## 5. Conclusions

Students are a particularly vulnerable population, and their vulnerability was aggravated by the COVID-19 pandemic. Several studies suggest that students’ mental health may continue to deteriorate even though the pandemic is now under control. Although this study has certain limitations which mean that the results should be treated with caution, our study was able to uncover novel findings that provide insight into the processes underlying students’ anxiety, depressive symptoms, and eating-related problems. In particular, it highlighted the central role of inhibitory IU in students’ anxiety and depressive symptoms, as well as the direct role of prospective IU in their eating-related problems. In this sense, our data suggest that IU should be a prime target for interventions to prevent and/or reduce the deterioration of students’ mental health. Future studies are needed to consolidate these findings now that the pandemic is under control, while diversifying data collection methods and designs to test these associations.

## Figures and Tables

**Figure 1 nutrients-15-04938-f001:**
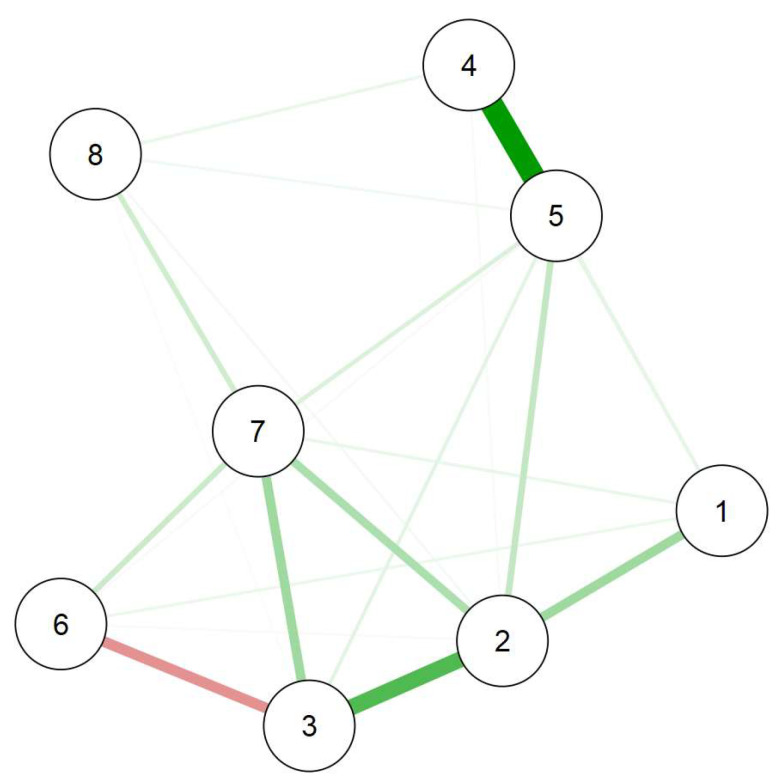
Network constructed using EBICglasso estimator. Note, 1 = fear of COVID-19; 2 = anxiety; 3 = depressive symptoms; 4 = prospective IU; 5 = inhibitory IU; 6 = adaptive coping; 7 = maladaptive coping; 8 = eating-related problems. The thickness of an edge reflects the magnitude of the association. Green lines represent positive correlations, and red lines negative correlations.

**Figure 2 nutrients-15-04938-f002:**
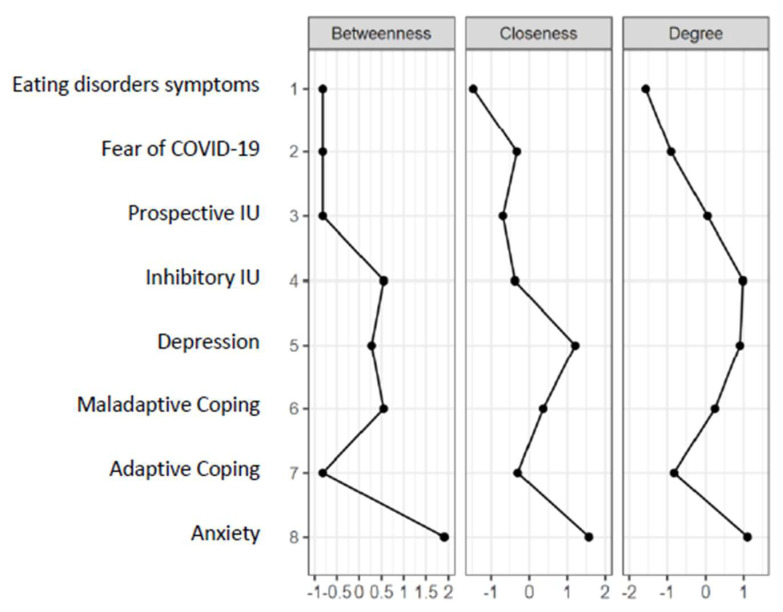
Centrality plots for EBICglasso network depicting the betweenness, closeness, and strength of each node.

**Figure 3 nutrients-15-04938-f003:**
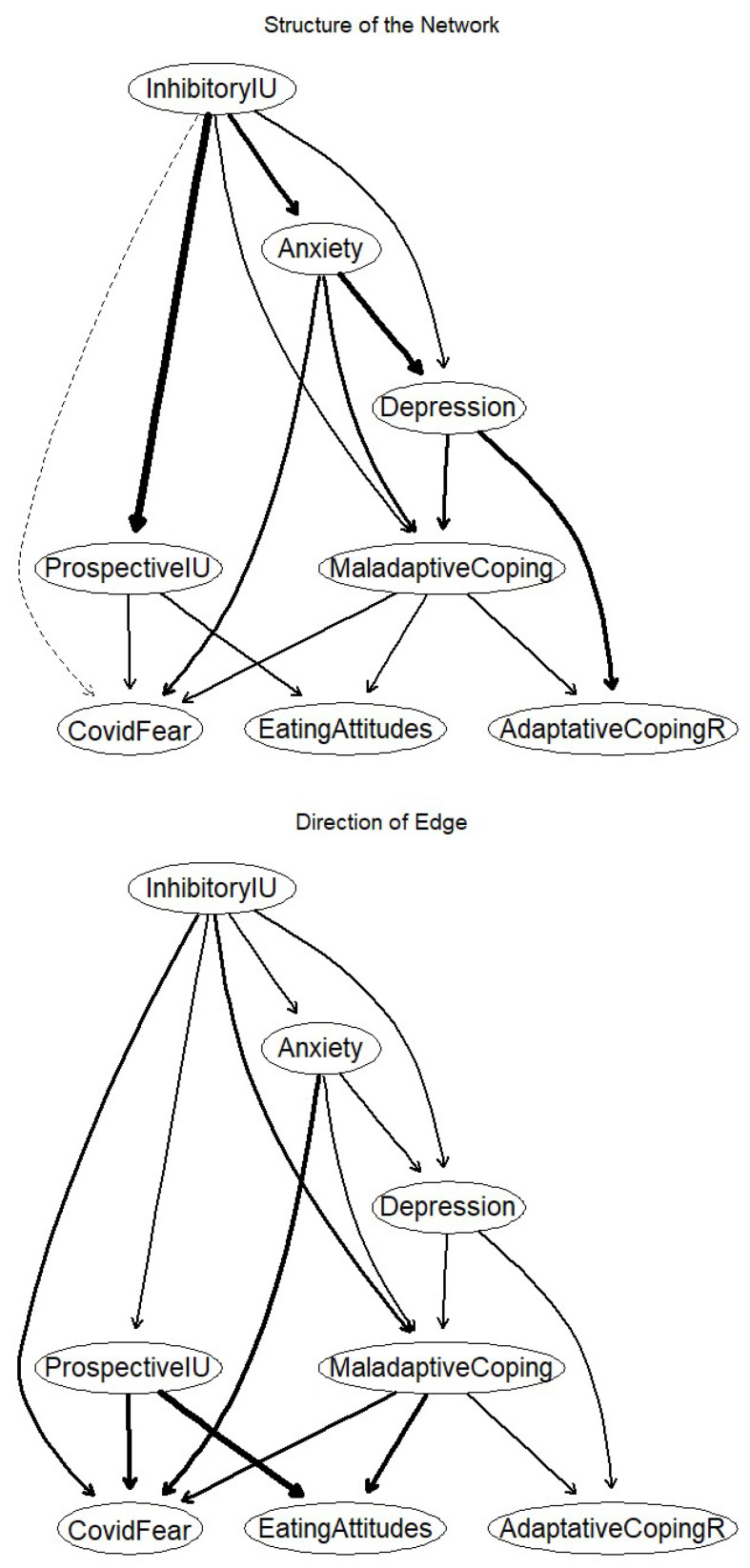
Directed acyclic graphs. Note: the network structure graph (represents the presence of edges (the thicker the line, the more powerful the association). Solid lines indicate edges that improved the model fit, while dashed lines indicate edges that worsened the model fit. The edge direction graph represents directional probability (the thicker the line, the greater the probability of the direction of the link). Solid lines represent edges that were present in their current direction in at least 51% of the 10,000 bootstrapped networks. InhibitoryIU = inhibitory intolerance of uncertainty; ProspectiveIU = prospective intolerance of uncertainty; AdaptiveCopingR = reversed adaptive coping score.

**Table 1 nutrients-15-04938-t001:** Participants’ descriptive data.

Major Disciplines	*n*	(%)
Psychology	514	(24)
Sports science	303	(14)
Law	175	(8)
Languages	148	(7)
Biology	143	(7)
Computer science	118	(6)
**Degree**		
First year	594	(28)
Second year	531	(25)
Third year	426	(20)
Professional degree	14	(1)
Fourth year	277	(13)
Fifth year	173	(8)
Doctorate	68	(3)
Other	56	(2)

**Table 2 nutrients-15-04938-t002:** Means, standard deviations, and Pearson’s correlation coefficients for our main variables of interest.

	*M*	*SD*	1		2		3		4		5		6		7		8	
1. Age	21.56	4.38	—															
2. Fear of COVID-19	14.35	5.86	0.06	**	—													
3. Anxiety	9.96	4.66	−0.01		0.41	***	—											
4. Depressive symptoms	7.60	4.29	0.02		0.28	***	0.63	***	—									
5. Intolerance of uncertainty (IU)	35.22	11.07	−0.10	***	0.31	***	0.44	***	0.35	***	—							
6. Prospective IU	21.55	6.74	−0.11	***	0.27	***	0.37	***	0.28	***	0.95	***	—					
7. Inhibitory IU	13.67	5.18	−0.07	**	0.31	***	0.47	***	0.39	***	0.91	***	0.72	***	—			
8. Eating-related problems	16.71	13.33	−0.04	*	0.10	***	0.18	***	0.16	***	0.20	***	0.18	***	0.19	***	—	
10. Adaptive coping	2.21	0.85	0.06	*	0.003		−0.11	***	−0.28	***	−0.09	***	−0.07	***	−0.10	***	−0.03	
11. Acceptance	3.53	1.66	−0.01		−0.24	***	−0.38	***	−0.41	***	−0.19	***	−0.15	***	−0.21	***	−0.09	***
12. Emotional support	2.59	1.83	−0.02		0.23	***	0.30	***	0.17	***	0.14	***	0.11	***	0.15	***	0.07	**
13. Humor	1.32	1.51	0.02		−0.28	***	−0.29	***	−0.31	***	−0.23	***	−0.21	***	−0.21	***	−0.10	***
14. Positive reframing	2.83	1.69	0.03		−0.17	***	−0.39	***	−0.46	***	−0.28	***	−0.24	***	−0.29	***	−0.11	***
15. Religion	1.09	1.74	0.05	*	0.16	***	0.09	***	0.11	***	0.06	**	0.04		0.08	***	0.03	
16. Active coping	1.84	1.47	0.09	***	0.04		−0.08	***	−0.24	***	−0.03		0.01		−0.08	***	−0.01	
17. Planning	2.27	1.65	0.06	**	0.03		−0.01		−0.17	***	0.01		0.05	*	−0.03		0.02	
18. Instrumental support	2.08	1.73	0.01		0.16	***	0.18	***	0.06	**	0.09	***	0.07	**	0.11	***	0.06	**
19. Venting	2.33	1.65	0.03		0.02		0.01		−0.12	***	−0.02		−0.03		−0.02		−0.02	
20. Maladaptive coping	1.98	0.95	−0.05	*	0.27	***	0.49	***	0.47	***	0.32	***	0.25	***	0.36	***	0.22	***
21. Denial	1.13	1.45	−0.03		0.24	***	0.23	***	0.24	***	0.20	***	0.17	***	0.21	***	0.13	***
22. Self-distraction	3.23	1.54	−0.06	**	0.10	***	0.02		−0.11	***	0.03		0.04		0.01		0.03	
23. Substance abuse	0.82	1.56	0.06	**	0.08	***	0.21	***	0.24	***	0.06	**	0.02		0.10	***	0.11	***
24. Behavioral disengagement	2.21	1.68	−0.05	*	0.18	***	0.38	***	0.47	***	0.27	***	0.21	***	0.29	***	0.15	***
25. Self-blame	2.53	1.88	−0.07	**	0.19	***	0.53	***	0.47	***	0.34	***	0.27	***	0.39	***	0.20	***

Note: * *p* < 0.05; ** *p* < 0.01; *** *p* < 0.001.

## Data Availability

The datasets generated during and/or analyzed during the current study are available from the corresponding author.

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
