# Peer review of "Relationship between Fear of COVID-19, Intolerance of Uncertainty, and Coping Strategies on University Students’ Mental Health"

_nutrients, 2023, doi:10.3390/nu15234938_

Round 1

Reviewer 1 Report

Comments and Suggestions for Authors

Thank you for the effort of the work presented the article showed the association between fear of COVID-19, intolerance of uncertainty, and coping strategies on university students' mental health. I thought that the paper was well-written, but I have the following minor comments.

1.    There is a difference in the prevalence of mental disorders between the sexes. The ratio should be added.

2.    Mental health would be strongly influenced by their own COVID-19 infection and their family death from COVID-19 infection. How background which could affect mental health did the participants have?

3. At page 1, the word of `Abstract` was duplicated.

Author Response

Dear reviewer 1, thank you for the opportunity to review our manuscript. At this point, we would like to express our thanks for strengthening our manuscript. In this response, we describe how we have addressed your concerns and recommendations. All changes in the revised version of the manuscript appear in green. We hope you will find that we have taken all your comments into account.

  1. There is a difference in the prevalence of mental disorders between the sexes. The ratio should be added.

Answer 1. Thank you for your suggestion. For each of our variables of interest, we performed a comparison of means (anova) to test for differences between women, men and non-binary people. Our results showed significant differences between these three groups. However, the effect sizes of these differences are very small. These results have been added to the manuscript in the results section, and the table has been added as an appendix given that these analyses, while important, are not strictly related to our hypotheses.

"In preliminary analyses, we studied the effect of gender on our different variables (Appendix 1). Although some statistically significant differences were present, the effect sizes of these differences were very small. More specifically, post hoc tests indicated that women experience slightly more fear of covid, depressive symptoms and eating-related problems than men. In addition, it appears that non-binary people experienced slightly more anxiety symptoms than women, who also experienced slightly more than men. Furthermore, women and non-binary people reported slightly higher intolerance of uncertainty and maladaptive coping than men.
Finally, there were no gender differences in adaptive coping.”

Appendix 1. Comparison of variables of interest by gender

Women (n=1612)

Men (n=506)

Nonbinary (n=21)

ANOVA

M

SD

M

SD

M

SD

F

p

η²

Fear of COVID

14.89

5.93

12.60

5.21

15.10

6.90

30.37

<.001

.03

Anxiety

10.45

4.56

8.25

4.55

13.33

4.48

50.75

<.001

.05

Depression

7.73

4.26

7.13

4.34

8.52

4.70

4.22

.02

.004

Eating-related problems

17.48

13.31

14.34

13.34

15.71

8.92

10.82

<.001

.01

Intolerance of uncertainty

36.08

11.20

32.39

10.14

38.86

12.07

22.91

<.001

.02

Prospective IU

21.95

6.84

20.28

6.29

22.24

6.20

12.07

<.001

.01

Inhibitory IU

14.13

5.19

12.11

4.79

16.62

5.90

33.46

<.001

.03

Adaptive coping

2.23

0.87

2.16

0.81

1.92

0.98

2.38

.09

.002

Maladaptive coping

2.04

0.94

1.77

0.92

2.32

0.98

17.98

<.001

.02

  1. Mental health would be strongly influenced by their own COVID-19 infection and their family death from COVID-19 infection. How background which could affect mental health did the participants have?

Answer 2. You're absolutely right, but we haven't evaluated these variables. We propose to add this point to the limitations of the study.

“Consequently, these results should be viewed with caution as a number of variables related to this specific context could have had an impact on some of our scores and were not captured in this study (e.g. having been directly or indirectly affected by COVID). For example, it has been shown that university students with COVID-19 symptoms, and/or having parents with COVID-19 symptoms, had higher levels of anxiety and depressive symptoms (4).”

  1. At page 1, the word of `Abstract` was duplicated.

Answer 3. Thank you, it's been fixed.

We remain at your disposal should you feel that further changes need to be made to the manuscript. 

Reviewer 2 Report

Comments and Suggestions for Authors

Dear colleagues!

I have read your work with great interest, but there are a number of questions and recommendations that I consider it appropriate to voice

1. Abstract

In line 9 you need to remove the repetition of the word “abstract”, and in line 15 - “results”

2. Keywords should be as concise as possible, so I recommend separating the phrases “inhibitory intolerance of uncertainty” and “prospective intolerance of uncertainty”, highlighting separately “inhibitory”, “prospective” and “intolerance of uncertainty”

3. In the introduction you have an empty line 52 - either remove it or fill it with the necessary text.

4. In hypotheses, you put 6 positions, but do not highlight zero. This section needs to be systematized.

5. Materials and methods

5.1 Survey

Unfortunately, I cannot agree with the reliability of the results that were obtained by analyzing questionnaires distributed through social networks, as the authors write. It is known that such a link can be used several times, and filling could also be done by one person.

In this regard, I want to know who and how validated the sociological survey method? Moreover, drawing conclusions about eating behavior based only on survey data is not far-sighted.

5.2 You write about non-binary participants on line 21. Did the sample size take into account this composition of respondents?

5.3. If you got the sample size estimate, why did you increase the number of respondents?

5.4. Again there is emptiness on lines 146-154. It looks like you didn't check the draft before uploading. This should be done to group the text more clearly.

6. Results

How can you evaluate hypotheses in general if you don't specify the null?

Сonclusions

Considering the low reliability of all survey tests conducted through social networks, I think your conclusions are overly bold. It is necessary to adhere to restrictions and formulate recommendations more precisely.

Author Response

Dear reviewer 2, thank you for the opportunity to review our manuscript. At this point, we would like to express our thanks for strengthening our manuscript. In this response, we describe how we have addressed your concerns and recommendations. All changes in the revised version of the manuscript appear in green. We hope you will find that we have taken all your comments into account.

  1. Abstract - In line 9 you need to remove the repetition of the word “abstract”, and in line 15 - “results”

Answer 1. Thank you, it's been fixed.

  1. Keywords should be as concise as possible, so I recommend separating the phrases “inhibitory intolerance of uncertainty” and “prospective intolerance of uncertainty”, highlighting separately “inhibitory”, “prospective” and “intolerance of uncertainty”

Answer 2. You're right, it's been fixed.

  1. In the introduction you have an empty line 52 - either remove it or fill it with the necessary text.

Answer 3. Thank you, it's been fixed.

  1. In hypotheses, you put 6 positions, but do not highlight zero. This section needs to be systematized.

Answer 4. Although we understand your concern, we have chosen this presentation because it conforms to the standards of writing and statistics in use in our field, psychology. Indeed, this approach enables us to present our hypotheses in a synthetic yet precise manner. We fear that the addition of null hypotheses will confuse the reader and make the manuscript more difficult to understand. In addition, although we have not written the null hypotheses corresponding to each of our hypotheses, highlighting the expected relationships seems to us to highlight the expected results. We hope you understand our position.

Furthermore, based on your comments, we realized that we had not always specified the correspondence between our results and our hypotheses. These have been systematically added to the manuscript, which seems to us to help clarify our purpose.

Materials and methods. 5.1 Survey - Unfortunately, I cannot agree with the reliability of the results that were obtained by analyzing questionnaires distributed through social networks, as the authors write. It is known that such a link can be used several times, and filling could also be done by one person. In this regard, I want to know who and how validated the sociological survey method? Moreover, drawing conclusions about eating behavior based only on survey data is not far-sighted.

Answer 5. First, concerning the reliability of the results that were obtained by analyzing questionnaires distributed through social networks, while we understand your reservations, during the lockdown, it seemed difficult to proceed differently. Supporting our point, many articles have been published using this type of data collection method during COVID-19 pandemic. For example (the first two examples were published in nutrients):  

Reviers et al., (2023): “Participants over 18 years of age were recruited through the researchers’ networks and through snowball techniques. Participants were invited to fill out the online questionnaire through a shared link.”

Reviers, A. D., Helme-Guizon, A., Moinard, C., & Féart, C. (2023). COVID-19 Lockdown and Changes in Dietary and Lifestyle Behaviors in a French Longitudinal Cohort. Nutrients, 15(21), 4682.

De Camargo et al., (2023): “Inclusion Criteria: adults (aged ≥18 years); Brazilians living in Brazil; acceptance of the informed consent document; access to social media, since the questionnaire was provided via digital communication platforms such as Facebook and Instagram. Participants who did not respond to the complete questionnaire were excluded from the sample. After electronic approval of the informed consent form, each participant had access to the “Questionnaire on physical activity habits, dietary and psychological aspects of lockdown due to the new coronavirus (COVID-19)”, available online on Google® Forms.”

de Camargo, E. M., López-Gil, J. F., Piola, T. S., Pechnicki dos Santos, L., de Borba, E. F., de Campos, W., & Gregorio da Silva, S. (2023). Association of the Practice of Physical Activity and Dietary Pattern with Psychological Distress before and during COVID-19 in Brazilian Adults. Nutrients, 15(8), 1926.

Wollast, et al., (2023): “Participants were recruited via Belgian online newspapers, mailing lists from universities, and social media platforms…”

Wollast, R., Preece, D. A., Schmitz, M., Bigot, A., Gross, J. J., & Luminet, O. (2023). The role of self-compassion in loneliness during the COVID-19 pandemic: a group-based trajectory modelling approach. Cognition and Emotion, 1-17.

Phillipou et al., (2020): “Briefly, the project includes a series of anonymous online surveys, open for 72 hr from the first to fourth of each month (for 12 months). After the initial 12 monthly surveys, annual surveys will be completed until 2024. Respondents were recruited through social media and other advertisements… »

Phillipou, A., Meyer, D., Neill, E., Tan, E. J., Toh, W. L., Van Rheenen, T. E., & Rossell, S. L. (2020). Eating and exercise behaviors in eating disorders and the general population during the COVID‐19 pandemic in Australia: Initial results from the COLLATE project. International Journal of eating disorders, 53(7), 1158-1165.

In addition, regardless of the specific context of the pandemic, which limited our possibilities for face-to-face data collection, the advantages and reliability of online data collection have been highlighted in several articles. Here are a few examples:

Regmi et al., (2016): “Data collection through an online survey appears to have the potential to collect large amounts of data efficiently (i.e. with less error due to the lack transferring written data on to a computer), economically (as it requires low human resource efforts while collecting or managing data) and within relatively short time frames. Online survey approach is also very useful when collecting data from hard-to-reach populations … Studying these sub-populations can be possible through an online survey approach as this may help access these hard to reach population by sending an invitation through a range of media and discussion platforms (e.g. social media, discussion fora).”

Regmi, P. R., Waithaka, E., Paudyal, A., Simkhada, P., & Van Teijlingen, E. (2016). Guide to the design and application of online questionnaire surveys. Nepal journal of epidemiology, 6(4), 640.

Van Selm & Jankowski (2006): “Other advantages to online surveys mentioned in the literature (Metha and Sivadas, 1995; Smith, 1997; Medlin et al., 1999; Brennan et al., 1999) include: absence of interviewer bias; removal of the need for data entry in as much as respondents directly enter data into an electronic file; convenience for respondents …. In summary, Sills and Song (2002) state that for particular populations that are “connected and technologically savvy”, the cost, ease, speed of delivery and response, ease of data cleaning and analysis weigh in favor of the Internet as a delivery method for survey research (Sills and Song, 2002: 28).”

Van Selm, M., & Jankowski, N. W. (2006). Conducting online surveys. Quality and quantity, 40, 435-456.

We hope that this information will be sufficient to convince you of the relevance of our data collection.

Second, in response to concerns about multiple use of the link, the settings of the platform we used, namely Qualtrics, give us the option of blocking multiple use of the link via the same terminal ("IP address recognition"). Furthermore, given that no compensation was offered to participants, we doubt they had any interest in proceeding in this way. Furthermore, we took care to eliminate questionnaires with aberrant answers (e.g. always giving the same answer to all questions) to ensure the reliability of responses. We suggest that you add this information to the manuscript to reassure readers of the reliability of our data collection.

“Qualtrics software ensures that a participant cannot respond twice to the link with the same IP address. In addition, all incomplete questionnaires and/or with aberrant answers (for example, with identical answers to all questions) have been deleted.”

Third, the study method was validated by the study's funder (Agence Nationale pour la Recherche) and its expert staff.

Finally, concerning your sentence “eating behavior based only on survey data is not far-sighted”, although the EAT 26 is considered a screening instrument for anorexia nervosa and bulimia nervosa, notably with a cut-off score of 20, we agree with you that, a simple self-evaluation scale is unreliable for making such a diagnosis. This is why we were careful to always use the term "problem" and not "disorder" throughout our manuscript when we talked about our results. Moreover, this scale is one of the most widely used standardized self-report measures for assessing the symptoms of eating problems and the risk of eating disorders in general, with good psychometric properties. To support our arguments, here's a link to over 100 references using EAT 26 in various populations: https://worldwidescience.org/topicpages/e/eating+attitude+test.html#

In addition, numerous articles have been published using this scale to measure problematic eating behaviors, using a method similar to ours, i.e. online data collection. For example:

Shokri et al., (2022): “This cross-sectional multicenter study was conducted through an online survey between March 22 and May 4, 2020. This period was the early phase of the COVID-19 outbreak in Iran when the Iranian government declared a national quarantine.

(…) Eating disorder symptoms and physical activity levels were evaluated using the Eating Attitudes Test-26 (EAT-26) and the International Physical Activity Questionnaire short form (IPAQ-SF), respectively.”

Shokri, F., Taheri, M., Irandoust, K., & Mirmoezzi, M. (2022). Effects of the COVID-19 pandemic on physical activity, mood status, and eating patterns of Iranian elite athletes and non-athletes. Zahedan Journal of Research in Medical Sciences, 24(3).

Martínez-de-Quel et al., (2021): Participants completed an anonymous online survey realized through Google Forms web survey platform. A personal invitation e-mail including the link to the web survey was sent via official channels of the involved universities. Besides, the survey was communicated through the social media (Facebook, LinkedIn and Twitter) and it was also shared to personal contacts of the researchers. The survey was hosted on the Google platform for a limited time window twice: two days after the state of emergency was issued (allotted time to submit questionnaire between March 16 and March 31, 2020), and five days after such measures began to be eased (between April 30 and May 11, 2020).

(…) The impact of the confinement on eating disorders was assessed by means of the Spanish version of the Eating Attitude Test-26 (EAT-26). The test includes 26 items related to bulimia, dieting and oral control, with a summative score ranging from 0 to 78. A cut-off at 20 was used to determine eating disorder cases from no-cases.

Martínez-de-Quel, Ó., Suárez-Iglesias, D., López-Flores, M., & Pérez, C. A. (2021). Physical activity, dietary habits and sleep quality before and during COVID-19 lockdown: A longitudinal study. Appetite, 158, 105019.

  • You write about non-binary participants on line 21. Did the sample size take into account this composition of respondents?

Answer. For each of our variables of interest, we performed a comparison of means (anova) to test for differences between women, men and non-binary people. Our results showed significant differences between these three groups. However, the effect sizes of these differences are very small which led us to include all participants, whatever their gender, in the rest of our analyses.. In addition, given that we had not hypothesized any gender differences, and that the results thus added highlight a very low incidence of gender, we did not explore this demographic variable further. These results have been added to the manuscript in the results section, and the table has been added as an appendix given that these analyses, while important, are not strictly related to our hypotheses.

"In preliminary analyses, we studied the effect of gender on our different variables (Appendix 1). Although some statistically significant differences were present, the effect sizes of these differences were very small. More specifically, post hoc tests indicated that women experience slightly more fear of covid, depressive symptoms and eating-related problems than men. In addition, it appears that non-binary people experienced slightly more anxiety symptoms than women, who also experienced slightly more than men. Furthermore, women and non-binary people reported slightly higher intolerance of uncertainty and maladaptive coping than men.
Finally, there were no gender differences in adaptive coping.”

Appendix 1. Comparison of variables of interest by gender

Women (n=1612)

Men (n=506)

Nonbinary (n=21)

ANOVA

M

SD

M

SD

M

SD

F

p

η²

Fear of COVID

14.89

5.93

12.60

5.21

15.10

6.90

30.37

<.001

.03

Anxiety

10.45

4.56

8.25

4.55

13.33

4.48

50.75

<.001

.05

Depression

7.73

4.26

7.13

4.34

8.52

4.70

4.22

.02

.004

Eating-related problems

17.48

13.31

14.34

13.34

15.71

8.92

10.82

<.001

.01

Intolerance of uncertainty

36.08

11.20

32.39

10.14

38.86

12.07

22.91

<.001

.02

Prospective IU

21.95

6.84

20.28

6.29

22.24

6.20

12.07

<.001

.01

Inhibitory IU

14.13

5.19

12.11

4.79

16.62

5.90

33.46

<.001

.03

Adaptive coping

2.23

0.87

2.16

0.81

1.92

0.98

2.38

.09

.002

Maladaptive coping

2.04

0.94

1.77

0.92

2.32

0.98

17.98

<.001

.02

5.3. If you got the sample size estimate, why did you increase the number of respondents?

Answer. In our view, it was important to estimate our sample size before starting our data collection, in order to set a target for our recruitment and ensure the validity of our analyses. During the recruitment phase, we recruited more participants because we had anticipated having to remove incomplete or aberrant responses. Once these had been removed, we ended up with more participants than we had anticipated. This was extremely positive for us, as it gave us greater statistical power for our analyses.

5.4. Again there is emptiness on lines 146-154. It looks like you didn't check the draft before uploading. This should be done to group the text more clearly.

Answer. Thank you, it's been fixed.

  1. Results How can you evaluate hypotheses in general if you don't specify the null?

Answer. As described above, we have chosen this presentation because it conforms to the standards of writing and statistics in use in our field, psychology. Indeed, this approach enables us to present our hypotheses in a synthetic yet precise manner. We fear that the addition of null hypotheses will confuse the reader and make the manuscript more difficult to understand. In addition, although we have not written the null hypotheses corresponding to each of our hypotheses, highlighting the expected relationships seems to us to highlight the expected results. Here are a few extracts from articles published in our field:

Deninotti et al., (2023): In accordance with the CSM, we made the following assumptions: (1) representations of infertility are associated with the adaptive and maladaptive coping strategies used by infertile people to deal with their infertility and (2) illness representations of infertility are related to the psychosocial outcomes of infertility (i.e., distress, anxiety, depressive symptoms, low well-being and poor quality of life).

Deninotti, J., Le Vigouroux, S., Gosling, C. J., & Charbonnier, E. (2023). Influence of illness representations on coping strategies and psychosocial outcomes of infertility: Systematic review and meta‐analysis. British Journal of Health Psychology.

Charbonnier et al., (2021): “Concerning our first objective, we hypothesized that during lockdowns, compared with periods after lockdown, university students exhibit more severe anxiety and depressive symptoms (H1), are more concerned about health (H2), and use more maladaptive strategies (e.g., behavioral disengagement) and fewer adaptive strategies (e.g., acceptance) (H3). Concerning our second objective, we hypothesized that the higher their levels of anxiety and depressive symptoms, the more concerned university students are about health (H4), and the more they use maladaptive strategies and the less they use adaptive strategies (H5).”

Charbonnier, E., Le Vigouroux, S., & Goncalves, A. (2021). Psychological vulnerability of French university students during the COVID-19 pandemic: a four-wave longitudinal survey. International journal of environmental research and public health, 18(18), 9699.

Whitney et al., (2007: “The aim of this study was to understand what contributes to the distress associated with caring for someone with an eating disorder. There were three main hypotheses. Firstly, carers would report high levels of distress. Secondly, greater negative and fewer positive appraisals of caregiving would be associated with higher levels of distress. Thirdly, carers’ own beliefs about the illness would be associated with carers’ distress, such that carers’ who perceive the illness to have greater consequences for the individual with an eating disorder and themselves, a longer timeline and little controllability by the person with an eating disorder would be experience higher levels of distress. In addition to predictors of carers’ distress, variables associated with carers’ positive and negative appraisals of caregiving were also examined.”

Whitney, J., Haigh, R., Weinman, J., & Treasure, J. (2007). Caring for people with eating disorders: Factors associated with psychological distress and negative caregiving appraisals in carers of people with eating disorders. British Journal of Clinical Psychology, 46(4), 413-428.

Pullmer et al., (2019): “The primary aim of this study was to elucidate the relations between self-compassion, psychological distress, and eating pathology. We hypothesized that (1) self-compassion would be negatively associated with psychological distress and eating pathology in female adolescents with eating disorders and (2) self-criticism would be positively associated with psychological distress and eating pathology. We also explored whether, as demonstrated in a school-based sample of adolescents (Pullmer et al., 2019b), psychological distress mediated the relation between self-compassion and eating pathology, as well as the relation between self-criticism and eating pathology.”

Pullmer, R., Zaitsoff, S. L., & Coelho, J. S. (2019). Self-compassion and eating pathology in female adolescents with eating disorders: The mediating role of psychological distress. Mindfulness, 10, 2716-2723.

Сonclusions - Considering the low reliability of all survey tests conducted through social networks, I think your conclusions are overly bold. It is necessary to adhere to restrictions and formulate recommendations more precisely.

Thank you for your caution. We have indeed moderated our conclusion section somewhat. However, we hope that our previous answers have reassured you of the confidence we can have in our results.

“Although this study has certain limitations which mean that the results should be treated with caution, our study was able to uncover novel findings that provide insight into the processes underlying students’ anxiety, depressive symptoms, and eating-related problems.”

“Future studies are needed to consolidate these findings now that the pandemic is under control, while diversifying data collection methods and study design to test these associations.”

-------

We remain at your disposal should you feel that further changes need to be made to the manuscript.

Reviewer 3 Report

Comments and Suggestions for Authors

I think the context of the study, the region or, if possible, the locality, should be detailed in the abstract.

In the method section, they should also make clear the number of French universities to which the survey was sent and the region or whether it was the entire country.

Author Response

Dear reviewer 3, thank you for the opportunity to review our manuscript. In this response, we describe how we have addressed your concerns and recommendations. We hope you will find that we have taken all your comments into account.

  1. I think the context of the study, the region or, if possible, the locality, should be detailed in the abstract.

Answer 1. This information has been added in abstract:

“Methods: 2139 French students of 54 university were recruited in the different regions of France during a French lockdown (between 21 April and 3 May 2021).”

  1. In the method section, they should also make clear the number of French universities to which the survey was sent and the region or whether it was the entire country.

Answer 2. You are right. This information has also been added:

“They came from 54 different universities across France, including the Universities of Paris City (21%) Nîmes (16%), Lorraine (14%), Montpellier (6%), and Strasbourg (5%)”.

Round 2

Reviewer 2 Report

Comments and Suggestions for Authors

Dear authors!

Now it has become obvious to me that your work was carried out at a high methodological level. Thank you, no other questions.

Author Response

Thank you for your positive feedback